# Physiological and Biochemical Responses and Transcriptome Analysis of *Bangia fuscopurpurea* (Rhodophyta) Under High-Temperature Stress

**DOI:** 10.3390/cimb47070484

**Published:** 2025-06-25

**Authors:** Minghao Zhao, Hongyan Zheng, Zepan Chen, Weizhou Chen

**Affiliations:** 1Marine Science Institute, Shantou University, Shantou 515063, China; 22mhzhao@stu.edu.cn (M.Z.); zhenghy@stu.edu.cn (H.Z.); zepan@stu.edu.cn (Z.C.); 2Guangdong Provincial Key Laboratory of Marine Biotechnology, Shantou University, Shantou 515063, China

**Keywords:** gene regulation, *Bangia fuscopurpurea*, macroalgae, transcriptome, photosynthesis, high-temperature stress

## Abstract

With the advancement of human industrial activities, increased carbon dioxide emissions have made global warming an inescapable trend. Elevated temperatures exert profound effects on the viability of large macroalgae. *Bangia fuscopurpurea* (Rhodophyta) is a commercially important large red alga widely cultivated along the coastal waters of Putian, Fujian Province, China; however, its physiological, biochemical, and molecular responses to heat stress remain unclear. To address this question, we cultured *B. fuscopurpurea* at 15 °C (control) and 28 °C (heat stress) for 7 days, assessed changes in growth and photosynthetic parameters, and performed transcriptome sequencing. Growth analysis revealed that the relative growth rate of *B. fuscopurpurea* at 28 °C was significantly lower than that at 15 °C. After 1 day at 28 °C, the chlorophyll a and carotenoid contents increased significantly; the phycobiliprotein levels rose markedly on days 4 and 7, whereas the *Fv/Fm* ratio decreased significantly on days 1, 4, and 7. Transcriptomic analysis indicated that heat stress up-regulated the majority of differentially expressed genes (DEGs) in *B. fuscopurpurea*. KEGG pathway enrichment analysis revealed that the DEGs were predominantly associated with photosynthesis, carbohydrate and energy metabolism, glycerophospholipid metabolism, and the glutathione cycle. In summary, *B. fuscopurpurea* mitigates the adverse effects of heat stress by up-regulating genes involved in photosynthesis, antioxidant defenses, and glycerophospholipid metabolism. These findings enhance our understanding of the physiological adaptations and molecular mechanisms by which *B. fuscopurpurea* responds to heat stress.

## 1. Introduction

Since the last century, anthropogenic emissions of greenhouse gases such as CO_2_ have increased markedly, leading to global climate change [1]. Studies predict that by 2100 the global mean sea surface temperature will increase by 3–7 °C [2]. As a crucial component of the Earth’s ecosystem, ocean warming will significantly affect the marine environment, including sea level rise [3], an increased frequency of marine heatwaves (MHWs) [4], and sea ice decline [5]. Macroalgae (including microalgae and large macroalgae), as the main primary producers of marine ecosystems, provide essential matter and energy for other consumers, playing an important role in marine ecosystems. In addition, they are responsible for approximately 50% of Earth’s oxygen production and biological carbon dioxide uptake, playing an important role in global oxygen supply, carbon dioxide fixation, and climate change mitigation [6]. The effects of ocean warming on the physiological and biochemical processes of large macroalgae indirectly affect their ecological functions and services in marine ecosystems. Studies have shown that large macroalgae can enhance their survival under high-temperature stress through physiological and molecular mechanisms [7]. High temperatures influence osmotic regulatory substances in macroalgae, such as soluble proteins and soluble sugars, as well as photosynthetic pigments [8]. Short-term high-temperature stress in *Saccharina japonica* triggers a series of physiological and biochemical responses, such as an increased total soluble protein content and enhanced antioxidant enzyme activities [9].

As integral components of marine ecosystems, intertidal macroalgae significantly influence oceanic carbon cycling and energy flux through photosynthesis. Under global climate change, thermal stress substantially affects the photosynthetic performance of macroalgae. In *Kappaphycus alvarezii*, photosynthetic performance and pigment content are significantly impaired by elevated temperatures, as evidenced by a reduction in the maximum photochemical quantum yield (Fv/Fm), indicating temperature sensitivity and damage to the photosynthetic apparatus [10]. Conversely, studies on Gracilaria lemaneiformis demonstrate that short-term thermal exposure stimulates growth and enhances photosynthetic activity [11], with photosynthetic saturation rates and carbon utilization efficiency both increasing with temperature [12]. Similar physiological responses have been documented in Gracilaria bailinae [7].

Bangia species are primarily distributed in the temperate and subtropical regions of the western North Pacific and the North Atlantic and are classified into marine and freshwater taxa [13]. *B. fuscopurpurea* predominantly grows on rocky substrates in the upper intertidal zone; it is a palatable, economically important red alga rich in EPA, free amino acids, polysaccharides, vitamins, and minerals, and among currently cultivated macroalgae, it has the highest EPA content, offering considerable commercial potential [14,15]. Currently, the artificial cultivation of red hair algae in China has been carried out only in Putian, Fujian [16]; however, the physiological responses and molecular mechanisms of marine Bangia thalli under high-temperature stress remain unclear. Therefore, this study investigates the effects of high-temperature stress on marine Bangia thalli in terms of growth and survival, physiological–biochemical parameters, and gene-level changes, aiming to provide further insight into the biological responses of marine Bangia to global warming and to facilitate the breeding of heat-tolerant strains.

## 2. Materials and Methods

### 2.1. Seaweed Samples and Temperature Treatment

*B. fuscopurpurea* samples were collected from aquaculture rafts at Nanri Island, Putian City, Fujian Province (25°13′17.4″ N, 119°28′31.9″ E). The thalli were air-dried and transported back to the laboratory on ice. The samples were first rinsed with seawater filtered through a 0.45 μm membrane to remove debris and epiphytes, then acclimated in an incubator for 2 days. The acclimation conditions were set to 15 °C, a light intensity of 40 μmol photons·m^−2^·s^−1^, pH 8.0, a 12 h light/12 h dark photoperiod, and a biomass density of 2 g·L^−1^; nutrients were supplied as NaNO_3_-N at 2 mg·L^−1^ and KH_2_PO_4_-P at 1 mg·L^−1^, with continuous aeration to maintain suspension and growth. After acclimation, the formal experiments were conducted.

Healthy and morphologically intact specimens of *B. fuscopurpurea* were cultured in 1 L of sterilized seawater medium at 15 °C and 28 °C, supplemented with 2 mg·L^−1^ NaNO_3_-N and 1 mg·L^−1^ KH_2_PO_4_-P. The culture conditions were as follows: a light intensity of 40 μmol photons m^−2^·s^−1^, seawater salinity of 30 psu, and a 12 h:12 h (light/dark) photoperiod. The experiment lasted 7 days with three replicates per group, and parallel samples were analyzed separately.

The culture conditions for the transcriptome materials were as follows: after the algae were returned to the laboratory and cultivated for two days as described above, they were transferred to sterile seawater containing antibiotics (0.1 g·L^−1^ kanamycin sulfate + 0.1 g·L^−1^ ampicillin + 0.2 g·L^−1^ streptomycin sulfate) for 1 day of cultivation, and then the formal experiments began. At 6 h and 1 d of the experiment, 3 samples of fresh *B. fuscopurpurea* with a weight of 0.3 to 0.5 g each were taken from each treatment group. These samples were rapidly frozen with liquid nitrogen and then stored at −80 °C for use as experimental materials for transcriptome sequencing.

### 2.2. Determination of Relative Growth Rate and Fluorescence Parameters

After high-temperature stress treatment, the fresh weight of *B. fuscopurpurea* was measured at 0, 1, 3, 5, and 7 days of high-temperature cultivation, and the relative growth rate was calculated using the formula proposed by Yong [17].

The maximum photochemical quantum yield of PSII (*Fv/Fm*) after t days of high-temperature stress was measured using IMAGING-PAM (MAXI, Walz, Nuremberg, Germany), with the formula *F_v_* = *F_m_* − *F*_0_. The minimal (*F*_0_) and maximal (*F_m_*) fluorescence was obtained by exposing the thalli, dark-adapted for 20 min, to actinic and saturating light provided by the fluorometer [18].

### 2.3. Measurement of Chlorophyll a and Carotenoids

Approximately 0.05 g of the algal sample was weighed into a 2 mL grinding tube, and 0.95 mL of anhydrous methanol was added. The sample was ground using a cryogenic grinder (JXFSTPRP-CLN-48, Shanghai Jingxin, Shanghai, China) at 70 Hz and −4 °C for 25 cycles (50 s grinding and 10 s pause per cycle). After grinding, the samples were left to stand at 4 °C for 24 h. The mixture was then centrifuged at 4000 rpm for 10 min at 4 °C. The supernatant was collected for further analysis. Using anhydrous methanol as a blank control, OD values at 480 nm, 510 nm, 652 nm, and 665 nm were measured with a microplate reader. The contents of chlorophyll a and carotenoids were calculated using the formula proposed by Porra [19].

### 2.4. Algal Phycobiliprotein Determination

Approximately 0.05 g (fresh weight) of algal tissue was placed in a 2 mL grinding tube, to which 0.95 mL of precooled 0.1 mol·L^−1^, pH 7.0 phosphate buffer was added. Samples were homogenized in a cryogenic grinder under the same conditions as described in Section 2.3. The homogenate was centrifuged at 4 °C and 4000 rpm for 10 min. The supernatant was collected for subsequent analysis. Using 0.1 mol·L^−1^, pH 7.0 phosphate buffer as the blank, OD values at 651 nm, 645 nm, 618 nm, 614 nm, 595 nm, 592 nm, 564 nm, and 455 nm were measured with a microplate reader (TCP011096, Jet Bio-Filtration, Guangzhou, China). The phycoerythrin and phycocyanin contents were calculated using the formula proposed by Beer [20].

### 2.5. Library Preparation for Transcriptome Sequencing

A total amount of 1 μg RNA per sample was used as input material for the RNA sample preparations. Sequencing libraries were generated using NEBNext^®^Ultra™ RNA Library Prep Kit for Illumina^®^ (NEB, USA) following the manufacturer’s recommendations, and index codes were added to attribute sequences to each sample. Briefly, mRNA was purified from total RNA using poly-T oligo-attached magnetic beads. Fragmentation was carried out using divalent cations under elevated temperature in NEBNext First Strand Synthesis Reaction Buffer (5×). First-strand cDNA was synthesized using a random hexamer primer and M-MuLV Reverse Transcriptase. Second-strand cDNA synthesis was subsequently performed using DNA polymerase I and RNase H. Remaining overhangs were converted into blunt ends via exonuclease/polymerase activities. After the adenylation of the 3′ ends of DNA fragments, NEBNext Adaptor with a hairpin-loop structure was ligated to prepare for hybridization. In order to select cDNA fragments of preferentially 240 bp in length, the library fragments were purified with the AMPure XP system (Beckman Coulter, Beverly, MA, USA). Then, 3 μL USER Enzyme (NEB, USA) was used with size-selected, adaptor-ligated cDNA at 37 °C for 15 min followed by 5 min at 95 °C before PCR. Then, PCR was performed with Phusion High-Fidelity DNA polymerase, Universal PCR primers, and Index (X) Primer. Finally, the library fragments were purified with AMPure XP system (Beckman Coulter, Beverly, USA), and the library quality was assessed with the Agilent Bioanalyzer 2100 system.

### 2.6. Quality Control, Transcriptome Assembly, and Gene Function Annotation

Raw data (raw reads) of fastq format were first processed through in-house perl scripts. In this step, clean data (clean reads) were obtained by removing reads containing the adapter, reads containing ploy-N, and low-quality reads from raw data. At the same time, the Q20, Q30, GC content, and sequence duplication level of the clean data were calculated. All the downstream analyses were based on clean data with high quality. The transcriptome was assembled using the Trinity software (2.14.0). Gene function was annotated based on the following databases: NR (NCBI non-redundant protein sequences); Pfam (Protein family); KOG/COG/eggNOG (Clusters of Orthologous Groups of proteins); Swiss-Prot (a manually annotated and reviewed protein sequence database); KEGG (Kyoto Encyclopedia of Genes and Genomes); and GO (Gene Ontology).

### 2.7. Analysis of Differentially Expressed Genes

The differential expression analysis of the two conditions/groups was performed using the DESeq R package (1.10.1). DESeq provides statistical routines for determining the differential expression in digital gene expression data using a model based on the negative binomial distribution. The resulting *p* values were adjusted using the Benjamini and Hochberg approach for controlling the false discovery rate. Genes with an adjusted *p* value < 0.05 found by DESeq were assigned as differentially expressed.

### 2.8. Statistical Analysis

The RGR, Chl-a, Car, PC, PE, and *Fv/Fm* data were expressed as means ± SD (n = 3). The experimental data were processed and statistically analyzed by Microsoft Office Excel software. The homogeneity of variance was tested by Levene’s test. At the significance level of *p* < 0.05, the statistical significance within each group at different time points was tested by One-Way ANOVA, and the statistical significance between groups at the same time point was tested by an independent-sample *T*-test. Graphpad prism 8.0 software was used for data plotting. Cluster trend profile chart, pairwise comparison, and volcano plot analysis was performed using BMKCloud (www.biocloud.net).

## 3. Results

### 3.1. Changes in Photosynthetic Pigments and Fluorescence Parameters Under Different Temperatures

*B. fuscopurpurea* exhibited different relative growth rates when cultured at 15 °C and 28 °C; from day 1 to day 7, the growth rate at 15 °C was consistently higher than at 28 °C, with significant differences (*p* < 0.05). At 15 °C, the maximum relative growth rate (6.95%) was observed on day 2, whereas at 28 °C, the peak rate (0.96%) occurred on day 4. The average relative growth rate was 5.6% at 15 °C and 0.59% at 28 °C (Figure 1).

The chlorophyll a and carotenoid contents of *B. fuscopurpurea* at 15 °C did not change significantly over time; at 28 °C, the chlorophyll and carotenoid levels initially declined, then increased, and subsequently declined again, reaching their lowest values at 6 h (0.86 and 0.21) and highest values at 1 d (1.21 and 0.29). Moreover, on day 1, the chlorophyll a and carotenoid levels in the 28 °C treatment (1.21 and 0.29) were significantly higher than those in the 15 °C treatment (0.94 and 0.23) (*p* < 0.05) (Figure 2A,B).

At 15 °C, the phycobiliprotein content in *B. fuscopurpurea* did not change significantly over time. At 28 °C, the phycobiliprotein content remained stable initially and then increased, with a significant rise from 3.59 on day 4 to 4.71 on day 7 (*p* < 0.05). Moreover, on days 1, 4, and 7, the phycobiliprotein contents at 28 °C (3.60, 3.59, 4.71) were significantly higher than those at 15 °C (3.06, 2.80, 3.48) (*p* < 0.05). Similarly, at 15 °C, the phycocyanin content in *B. fuscopurpurea* did not exhibit significant changes over time. At 28 °C, the phycocyanin content first decreased and then increased, with a significant rise from 0.37 at 6 h to 0.61 on day 4 (*p* < 0.05) and further to 1.06 on day 7 compared to 0.61 on day 4 (*p* < 0.05). Additionally, on days 4 and 7, the phycocyanin contents in the 28 °C group (0.61, 1.06) were significantly higher than those in the 15 °C group (0.35, 0.50) (*p* < 0.05) (Figure 2C,D). At 15 °C, the Fv/Fm of *B. fuscopurpurea* exhibited an initial increase followed by stabilization: at 6 h, the Fv/Fm (0.48) was significantly higher than at 0 d (0.44) (*p* < 0.05) and thereafter showed no significant change over time. At 28 °C, the *Fv/Fm* showed a rise-and-fall pattern: at 1 d (0.51), it was significantly higher than at 6 h (0.42) (*p* < 0.05), then declined significantly to 0.46 at 4 d compared to 1 d (0.51) and further to 0.38 at 7 d compared to 4 d (0.46) (*p* < 0.05). Moreover, at 28 °C, the *Fv/Fm* values at 6 h, 4 d, and 7 d were 0.42, 0.46, and 0.38, respectively, which were significantly lower than the corresponding values at 15 °C (0.48, 0.51, and 0.50) (Figure 3).

### 3.2. Data Quality

The transcriptome sequencing of twelve samples generated a total of 77.97 Gb of clean data, with each sample yielding at least 5.97 Gb and a Q30 base percentage of ≥85.50% (Appendix A). Assembly produced 47,609 unigenes in total. Of these, 6945 unigenes exceeded 1 kb in length. The functional annotation of the unigenes yielded annotations for 22,000 entries (Appendix A).

### 3.3. Gene Function Annotation

Currently, the whole-genome sequencing of *B. fuscopurpurea* remains incomplete; therefore, unigene sequences were compared against the NR, Swiss-Prot, COG, KOG, eggNOG4.5, and KEGG databases using DIAMOND (v2.0.4). KEGG Orthology assignments were obtained via KOBAS (v2.0), and GO Orthology annotations were derived by analyzing predicted gene sequences with InterProScan (5.34-73.0) against the integrated InterPro database. Following amino acid prediction, HMMER (v3.1b2) searches against the Pfam database were performed to retrieve comprehensive unigene annotation information. By applying BLAST with an E-value cutoff of ≤1 × 10^−5^ and HMMER with an E-value threshold of ≤1 × 10^−10^, this study ultimately obtained annotation information for 22,000 unigenes.

In the NR database, the top match was *Porphyra umbilicalis* (54.87%), which, like *B. fuscopurpurea*, belongs to the Bangiaceae family, consistent with their taxonomic relationship (Appendix A). A total of 7701 genes were annotated in the COG database and grouped into 25 functional categories: the largest was “Translation, ribosomal structure and biogenesis” (1172), followed by “Posttranslational modification, protein turnover, and chaperones” (880), and the smallest was “RNA processing and modification” (3) (Appendix A). The GO database annotated 14,577 genes, which were distributed among three main categories: Cellular Component, Molecular Function, and Biological Process. Within the Cellular Component category, the largest term was “cellular anatomical entity,” followed by “intracellular” and then “protein complex.” In the Molecular Function category, “binding” was predominant, followed by “catalytic activity” and then “structural molecule activity.” In Biological Process, “cellular process” was the most represented, followed by “metabolic process” and then “biological regulation” (Appendix A). The number of genes annotated across all databases was 3397 (Appendix A).

### 3.4. Differential Gene Analysis

#### 3.4.1. Overview of Differentially Expressed Genes

Using log_2_(FC) ≥ 2 and FDR < 0.01 as thresholds for differential gene screening, two comparisons were established, CT-6 h vs. HT-6 h and CT-1 d vs. HT-1 d, to investigate temporal changes in gene expression under heat stress. A total of 3644 DEGs were identified across time points: 1677 DEGs (979 up-regulated and 698 down-regulated) in CT-6 h vs. HT-6 h and 1967 DEGs (1261 up-regulated and 706 down-regulated) in CT-1 d vs. HT-1 d (Figure 4A,B). Among these, 813 DEGs were modulated by heat treatment at both 6 h and 1 d (Figure 4C). In addition, the trend analysis of DEG expression over time under heat stress in *B. fuscopurpurea* yielded eight clustered expression profiles (Figure 5). Profiles A and D (122 and 347 DEGs, respectively) showed expression up-regulation over time at both control and elevated temperatures. Clusters B and E (432 and 647 DEGs, respectively) were characterized by high expression under heat stress and low expression at the control temperature. Clusters C and F (78 and 368 DEGs, respectively) displayed increasing expression over time under heat stress, with minimal change at the control temperature.

#### 3.4.2. Analysis of Differentially Expressed Genes Using GO and KEGG Databases

GO and KEGG functional annotations were performed for the DEGs. GO classification revealed that DEGs are involved in three categories, Biological Process, Molecular Function, and Cellular Component, and in both CT-6 h vs. HT-6 h and CT-1 d vs. HT-1 d comparisons, most genes were specifically assigned to metabolic processes, cellular processes, and biological regulation (Appendix A). KEGG enrichment analysis showed that at 6 h DEGs were primarily enriched in carbon metabolism, amino acid biosynthesis, and protein processing in the endoplasmic reticulum; at 1 d, they were mainly enriched in ribosome, carbon metabolism, and amino acid biosynthesis (Appendix A and Table 1).

#### 3.4.3. KEGG Differential Expression Pathway Analysis

Based on the KEGG enrichment of DEGs, eight pathways related to energy, carbohydrate, and lipid metabolism were selected to examine the temporal gene expression in *B. fuscopurpurea* under heat stress: photosynthesis (ko00195), photosynthesis antenna proteins (ko00196), carbon fixation in photosynthetic organisms (ko00710), glycolysis/gluconeogenesis (ko00010), the pentose phosphate pathway (ko00030), the Citrate cycle (ko00020), glycerophospholipid metabolism (ko00564), and glutathione metabolism (ko00480). The full gene names used in all heatmaps of DEGs are provided in Appendix A.

##### Photosynthesis Pathway

Comparative transcriptome analysis identified 13 DEGs involved in photosynthesis, participating in the assembly of PSII (*PsbA*, *PsbC*, *PsbM*, *PsbO*, *PsbP*, *PsbQ*, *PsbS*, *PsbU*), PSI (*PsaO*), the cytochrome b6/f complex (*PetC*), and photosynthetic electron transport (*PetF*, *PetH*) (Figure 6). All of these DEGs were significantly up-regulated at both 6 h and 1 d, with the two down-regulated genes (*PsbA*, *PsbC*) being enriched at 1 d. In addition, five DEGs related to light-harvesting antenna proteins were identified, involved in the synthesis of the allophycocyanin β subunit (*ApcB*), phycobilisome linker protein (*ApcC*), phycocyanin rod-linker protein (*CpcC*), phycobilin lyase α subunit (*CpcE*), and photosystem I light-harvesting complex (*Lhca1*). All of these antenna-related DEGs were significantly up-regulated at both 6 h and 1 d, with the single down-regulated gene (*ApcB*) enriched at 1 d.

**Figure 6 cimb-47-00484-f006:**
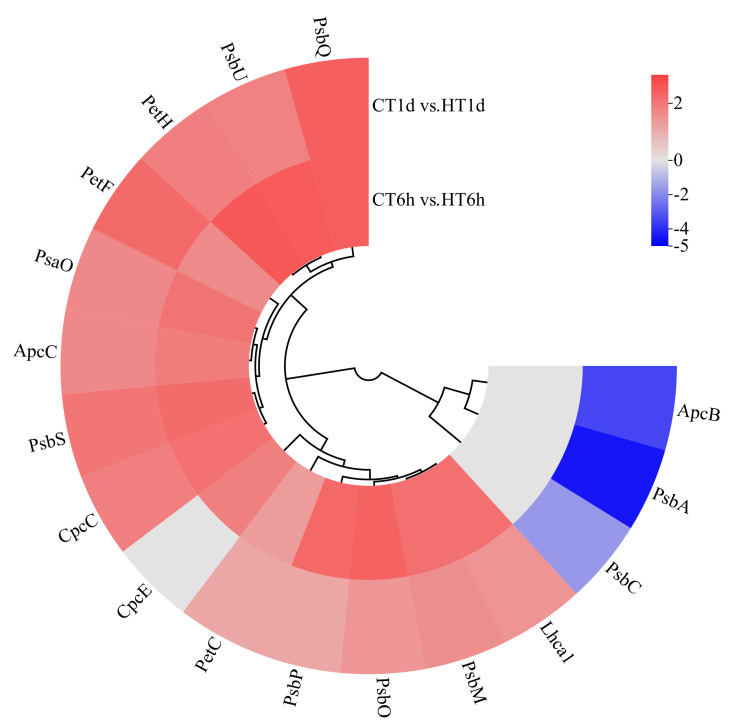
Cluster heatmap of differentially expressed genes in photosynthesis. Note: The logarithmic values of fold change (Log_2_FC) of expression levels among the sample groups are shown in the legend, where red indicates significant up-regulation, blue indicates significant down-regulation, and gray indicates no significant difference.

##### Carbohydrate Synthesis and Energy Metabolism Pathways

Transcriptome analysis revealed that DEGs are significantly enriched in pathways related to carbon and energy metabolism and play roles in the temporal response of *B. fuscopurpurea* to heat stress. In our findings, the Calvin cycle was significantly enriched with 10 DEGs: *SBPase* was uniquely up-regulated at 6 h; *RpiA*, *TK*, *TPI*, and *MDH2* were exclusively up-regulated at 1 d; *RbcL* was down-regulated at 1 d; and *ALDO*, *FBP*, *PrkB*, and *RPE* were significantly up-regulated at both time points (Figure 7A). The Calvin cycle intermediates glyceraldehyde-3-phosphate and dihydroxyacetone phosphate can funnel into glycolysis/gluconeogenesis for sucrose and starch synthesis, enter the tricarboxylic acid cycle to generate energy, or flow into the pentose phosphate pathway to produce NADPH. In this study, glycolysis/gluconeogenesis was significantly enriched with nine DEGs. *TPI*, *FrmA*, and ADH were exclusively up-regulated at 1 d; *PK* was significantly down-regulated at both time points; and *FBP*, *Pgm*, *ALDO*, *PdhD*, and *G6PE* were significantly up-regulated at both 6 h and 1 d (Figure 7B). The pentose phosphate pathway was significantly enriched with nine DEGs: *PGD* was down-regulated solely at 6 h; *RpiA*, *G6PD*, *PGLS*, and *TK* were up-regulated at both time points; and *ALDO*, *FBP*, *Pgm*, and *RPE* were significantly up-regulated at both 6 h and 1 d (Figure 7C). The TCA cycle was significantly enriched with seven DEGs: *SucA* was exclusively up-regulated at 6 h; *CS*, *ACO*, *FumC*, and *MDH2* were up-regulated exclusively at 1 d; *SucB* was down-regulated at 1 d; and *PdhD* was significantly up-regulated at both time points (Figure 7D).

##### Glycerophospholipid Metabolism Pathway

Transcriptome analysis revealed the enrichment of eight DEGs involved in glycerophospholipid metabolism, encoding enzymes for glycerol-3-phosphate dehydrogenase (*GPD1*), another glycerol-3-phosphate dehydrogenase isoform (*GlpA*), diacylglycerol kinase (*DgkA*), acylglycerophosphate acyltransferase 8 (*AGPAT8*), phosphatidylethanolamine N-methyltransferase (*PEMT*), ethanolamine-phosphate transferase (*EPT1*), cytidine diphosphate–ethanolamine synthase (*PCYT2*), and phosphatidylserine decarboxylase (*PSD*). Specifically, *GPD1* and *GlpA* were exclusively down-regulated at 6 h; *PSD* and *PCYT2* showed exclusive up-regulation at 1 d; *AGPAT8*, *DgkA*, and *EPT1* were significantly up-regulated at both time points; and *PEMT* exhibited down-regulation at 6 h followed by up-regulation at 1 d (Figure 8).

##### Glutathione Metabolism Pathway

Transcriptome analysis revealed that heat stress induced the differential expression of nine DEGs involved in the glutathione cycle, encoding enzymes such as leucine aminopeptidase (*PepA*), γ-glutamyl transpeptidase (*Ggt*), 6-phosphogluconate dehydrogenase (*PGD*), glutathione peroxidase (*Gpx*), ornithine decarboxylase (*ODC*), glutathione S-transferase (*HPGDS*), glucose-6-phosphate dehydrogenase (*G6PD*), L-ascorbate peroxidase (*APX*), and ribonucleotide reductase subunit M2 (*RRM2*). Among these, *Ggt* was uniquely up-regulated at 6 h; *PepA*, *PGD*, and *ODC* were exclusively down-regulated at 6 h; *G6PD* was specifically up-regulated at 1 d; and *HPGDS*, *Gpx*, *RRM2*, and *APX* were significantly up-regulated at both time points (Figure 9).

## 4. Discussion

As global warming intensifies, large-scale economically cultivated macroalgae are inevitably subjected to the adverse effects of elevated temperatures. Our results demonstrate that *B. fuscopurpurea* grows more favorably at 15 °C compared to at 28 °C. Short-term heat exposure (1 day) significantly increased the chlorophyll a and carotenoid contents while markedly reducing the *Fv/Fm*. Transcriptomic analysis uncovered the molecular mechanisms underlying heat tolerance in this alga. RNA-seq data indicate that these responses involve the regulation of carbohydrate metabolism, photosynthesis, glycerophospholipid metabolism, and the glutathione cycle.

### 4.1. The Influence of Temperature on Photosynthetic Pigments and Fluorescence Parameters

Photosynthesis and respiration in macroalgae are temperature-dependent, and fluctuations in temperature can alter algal growth performance [21]. Different algal taxa exhibit variable tolerance to temperature shifts, and the growth rate serves as a key metric that directly reflects their physiological condition [22]. Under heat stress, the Atlantic brown alga Fucus vesiculosus exhibited a 65% reduction in growth rate at 26 °C compared to at 12 °C [23]. Similarly, in our study, Bangia fuscopurpurea cultured at 28 °C displayed a significantly suppressed relative growth rate compared to that cultured at 15 °C. Chlorophyll a, the principal light-harvesting pigment, directly determines photosynthetic capacity and indirectly influences the growth rate; carotenoids protect chlorophyll from photodamage, and phycobiliproteins capture and transfer light energy to chlorophyll, all playing crucial roles in photosynthesis [24]. Studies have shown that under heat stress, *Ulva prolifera* attains peak chlorophyll a content after 4 days of cultivation [25]. Likewise, *Gracilaria blodgettii* shows significant increases in chlorophyll a and carotenoid levels under elevated temperatures, which then decline as the temperature rises further [26]. Phycoerythrin (PE), a major light-harvesting pigment found in red algae and cyanobacteria [27], also exhibits potent antioxidant activity by scavenging reactive oxygen species and alleviating oxidative stress [28]. In prolonged-heat-stress experiments with *Neopyropia yezoensis*, both the phycoerythrin and phycocyanin contents were found to increase significantly [29]. Data analysis revealed that in the 28 °C treatment, the chlorophyll a and carotenoid levels in *B. fuscopurpurea* decreased at 6 h and rose at 1 day (*p* < 0.05), while phycoerythrin and phycocyanin exhibited a progressive increase, significantly peaking on day 7 (*p* < 0.05), consistent with previous studies. Photosystem II (PSII) is regarded as one of the most temperature-sensitive components of the photosynthetic apparatus [30,31]. The *Fv/Fm* ratio represents the maximum quantum yield of PSII, reflecting the alga’s maximal photosynthetic efficiency and indirectly indicating its growth status [7]. Studies have shown the Fv/Fm to be a valuable metric for assessing macroalgal thermal tolerance [32]. Our results demonstrate that the *Fv/Fm* values in *B. fuscopurpurea* at 28 °C at 6 h, 4 d, and 7 d were significantly lower than those at 15 °C (*p* < 0.05), indicating initial photoinhibition at 6 h, consistent with the findings in *Kappaphycus alvarezii* [33]. Over the extended exposure periods (4 d and 7 d), heat stress progressively impaired the photosynthetic apparatus, further reducing the *Fv/Fm* (*p* < 0.05).

### 4.2. The Influence of Temperature on the Photosynthetic Pathway

Photosynthesis is one of the most temperature-sensitive biological processes in algae [34], relying on electron transport within the photosystems to supply the energy required for algal growth and development [35]. In red algae, phycobilisomes serve as the principal light-harvesting antennae in the photosynthetic mechanism. Red algal phycobilisomes consist of phycoerythrin (PE), phycocyanin (PC), and allophycocyanin (APC); the red coloration of these algae arises from APC masking the green color of chlorophyll and other pigments [36]. These phycobilisome structures are closely associated with Photosystem II (PSII) and Photosystem I (PSI), ensuring the efficient transfer of excitation energy to their reaction centers and promoting the conversion of light into chemical energy [37]. Studies have shown that in Sargassum horneri, the expression of photosynthesis-related genes is down-regulated during the early phase of heat stress [38]. In this study, genes encoding the PSII dimer subunit PsbM and oxygen-evolving enhancer proteins (*PsbO*/*P*/*Q*/*U*) were rapidly up-regulated to stabilize the oxygen-evolving complex and maintain water-splitting activity, ensuring sustained ATP and NADPH production. Concurrently, genes for photoprotective proteins such as PsbS and Lhca1, along with key electron transport components (*PetC*/*F*/*H*), were co-up-regulated, balancing the energy flow between PSII and PSI, preserving non-photochemical quenching and thermal dissipation, and preventing excessive ROS accumulation. At 6 h, the up-regulation of *CpcE* likely reinforces phycobilisome integrity, followed by the moderate down-regulation of core reaction-center proteins (*PsbA*/*C*, *ApcB*) to reduce light capture and mitigate photoinhibition risk—findings consistent with observations in *Pyropia haitanensis* under heat stress [39].

### 4.3. The Influence of Temperature on Carbohydrate Synthesis and Energy Metabolism Pathways

Studies have shown that algal metabolic processes are subject to gene-level regulation under stress conditions [40], with carbohydrate metabolism representing a critical component. Algae modulate carbohydrate concentrations and structural composition to optimize the use of endogenous carbon-derived energy and accumulate compatible solutes for molecular protection [41]. However, under excessive environmental stress, algae may experience energy deficits, prompting the up-regulation of intrinsic carbohydrate metabolic routes and the activation of alternative pathways such as glycolysis to sustain ATP supply and carbon skeleton provision for essential processes [42]. For example, *G. lemaneiformis* exhibits a significant accumulation of floridoside and isofloridoside after 1 and 2 days of heat stress [43]. In this study, we conducted an integrated analysis of four carbohydrate and energy metabolism pathways: the Calvin cycle, glycolysis/gluconeogenesis, the pentose phosphate pathway, and the tricarboxylic acid (TCA) cycle. We observed that the majority of Calvin cycle genes were up-regulated, consistent with the response reported in *Saccharina latissima* under elevated temperature [44]. However, *RbcL* was significantly down-regulated at 1 day; the differential temperature sensitivity of Rubisco has been reported across algal taxa [45], and our finding aligns with that of Huang, who observed *RbcS* down-regulation in *G. bailinae* under heat stress [7]. Previous studies suggest that the thermal lability of the electron transport chain may constrain energy metabolism under heat stress [46]. For instance, *P. haitanensis* shows a significant down-regulation of energy metabolism pathways, including the pentose phosphate pathway and TCA cycle, under heat stress [39], whereas *G. lemaneiformis* displays enhanced energy metabolism when exposed to high temperature, mirroring our observations [47]. The enhanced pentose phosphate pathway and glycolysis/gluconeogenesis generate abundant NADPH, supplying energy and supporting antioxidant defenses by maintaining enzyme activities to mitigate oxidative damage. While heat stress has been reported to inhibit the TCA cycle in *Sargassum fusiforme* [48], our study found an overall up-regulation of TCA cycle genes. Key intermediate-synthesizing enzymes (*CS*, *MDH2*, *PdhD*) were significantly induced, indicating that enhanced TCA activity under heat stress bolsters signal transduction, energy supply, and antioxidant defense to counteract thermal damage.

### 4.4. The Influence of Temperature on Glycerophospholipid Metabolism

Glycerophospholipids, as the principal lipid constituents of biological membranes, are essential for maintaining bilayer fluidity, a property fundamental to the proper function of membrane-bound proteins, ion channels, and receptors [49]. Under heat-stress conditions, the lipid composition and architecture of algal plasma membranes may be altered [50]; the regulation of glycerophospholipid metabolism is therefore critical for preserving membrane fluidity and integrity, which in turn protects cells from thermal damage. Studies have shown that chitosan oligosaccharide treatment enhances thermotolerance in Gracilariopsis lemaneiformis by significantly up-regulating genes involved in glycerophospholipid metabolism, increasing photosynthetic membrane lipid content, and promoting the accumulation of lipid signaling molecules such as phosphatidic acid (PA), thereby improving photosynthetic growth and heat resistance under high-temperature stress [51]. Similarly, exogenous arginine application in heat-stressed Sargassum fusiforme induces the up-regulation of arginine metabolism genes, leading to enhanced phosphatidic acid synthesis and improved thermotolerance [52]. Consistent with these findings, our data show that at 6 h of heat stress, *B. fuscopurpurea* limits excessive lipid synthesis or redirects carbon flux by reducing glycerol-3-phosphate production, while the sustained up-regulation of *DgkA* promotes the conversion of diacylglycerol (DAG) to phosphatidic acid (PA). PA not only fuels phospholipid biosynthesis but also acts as a signaling molecule to orchestrate membrane remodeling and stress-response pathways, facilitating rapid membrane repair and mitigating heat-induced cellular damage.

### 4.5. The Influence of Temperature on Glutathione Metabolism

Previous studies have demonstrated that plants deploy specific genetic responses to cope with environmental stresses, with glutathione (GSH) and hydrogen peroxide serving as central signaling molecules in both abiotic and biotic stress pathways [53]. The exogenous application of GSH can enhance antioxidant enzyme activities, mitigate oxidative damage, and improve thermotolerance in plants [54]. In macroalgae, the principal mechanism for counteracting excessive reactive oxygen species is the ascorbate–glutathione cycle [55]. Maintaining the dynamic equilibrium between reduced glutathione (GSH) and its oxidized form (GSSG), as well as between ascorbate (ASC) and dehydroascorbate (DHA), is essential for alleviating oxidative stress. Moreover, the activation of antioxidant enzymes—such as glutathione reductase (GR), ascorbate peroxidase (APX), and superoxide dismutase (SOD)—also plays a pivotal role in this defense system [56]. Previous research has shown that the ascorbate–glutathione cycle participates in the low-salinity stress response of *B. fuscopurpurea* [57]. Similarly, in our study, *B. fuscopurpurea* exhibited a significant up-regulation of the γ-glutamyl transpeptidase gene (*Ggt*) at 6 h of heat stress, enhancing GSH synthesis to rapidly scavenge intracellular ROS and ameliorate acute oxidative damage. After 1 day of stress, the thalli had reprogrammed their metabolic networks and engaged additional heat-response pathways, resulting in a normalization of *Ggt* expression. Throughout heat stress, glutathione peroxidase (*GPX*) and L-ascorbate peroxidase (*APX*) enzymes directly detoxifying H_2_O_2_ and other ROS must remain highly active; accordingly, *GPX* and *APX* genes were significantly up-regulated at both 6 h and 1 d to sustain cellular homeostasis.

### 4.6. The Gene Regulatory Mechanism Under High-Temperature Stress

Integrating the foregoing pathway analyses, it can be concluded that under heat stress, *B. fuscopurpurea* orchestrates a complex adaptive response network via precise gene regulation. Transcriptome profiling indicates that heat stress induces the modulation of genes involved in energy transfer between PSII and PSI, light harvesting, and heat-shock proteins, thereby mitigating photodamage. Concurrently, it fine-tunes genes in the Calvin cycle, the TCA cycle, glycolysis/gluconeogenesis, and the pentose phosphate pathway to balance ATP production with carbon fixation while triggering antioxidant defenses. In glycerophospholipid metabolism, the sustained up-regulation of *DgkA* drives the conversion of diacylglycerol into phosphatidic acid, promoting membrane remodeling and repair. Moreover, within 6 h of heat exposure, *Ggt* is rapidly induced to sustain the glutathione cycle, followed by the persistent up-regulation of *GPX* and *APX* to scavenge ROS, collectively preserving cellular function and viability. Together, these coordinated regulatory events constitute the comprehensive response strategy of *B. fuscopurpurea* under heat stress (Figure 10).

## 5. Conclusions

In summary, this study conducted a preliminary investigation into the physiological and molecular mechanisms of *B. fuscopurpurea* under heat stress. Cultivation at 28 °C imposed thermal stress on *B. fuscopurpurea*, resulting in a significant reduction in the relative growth rate and *Fv/Fm* values. The chlorophyll a and carotenoid contents significantly increased on day 1, while the phycobiliprotein levels showed a marked rise on days 4 and 7. Transcriptomic analysis revealed that *B. fuscopurpurea* adapts to high-temperature stress by regulating eight metabolic pathways related to photosynthesis, energy and carbohydrate metabolism, glycerophospholipid metabolism, and the glutathione cycle. The alga mitigates heat-induced damage by enhancing the Calvin cycle, the TCA cycle, and glycolysis/gluconeogenesis pathways, thereby accelerating the synthesis of osmotic regulators and NADPH. Furthermore, this study found that *B. fuscopurpurea* may counteract the adverse effects of heat stress by up-regulating genes involved in glycerophospholipid and glutathione metabolism to alter membrane fluidity and improve the reactive oxygen species scavenging capacity. These findings provide a valuable reference for further research on stress resistance and the development of heat-tolerant cultivars of *B. fuscopurpurea*.

## Figures and Tables

**Figure 1 cimb-47-00484-f001:**
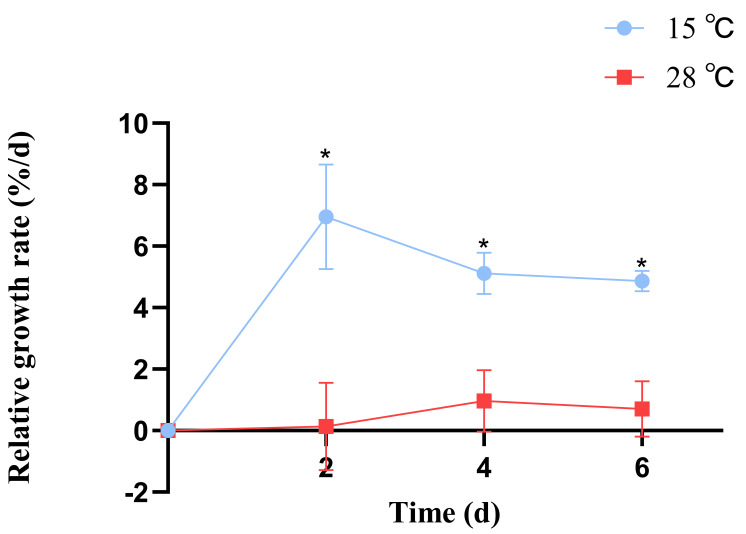
The influence of different temperatures on the relative growth rate of *B. fuscopurpurea*. Note: * means that there is a significant difference between the 15 °C and 28 °C groups at the same time point; * means that there is a statistical difference at the *p <* 0.05 significance level.

**Figure 2 cimb-47-00484-f002:**
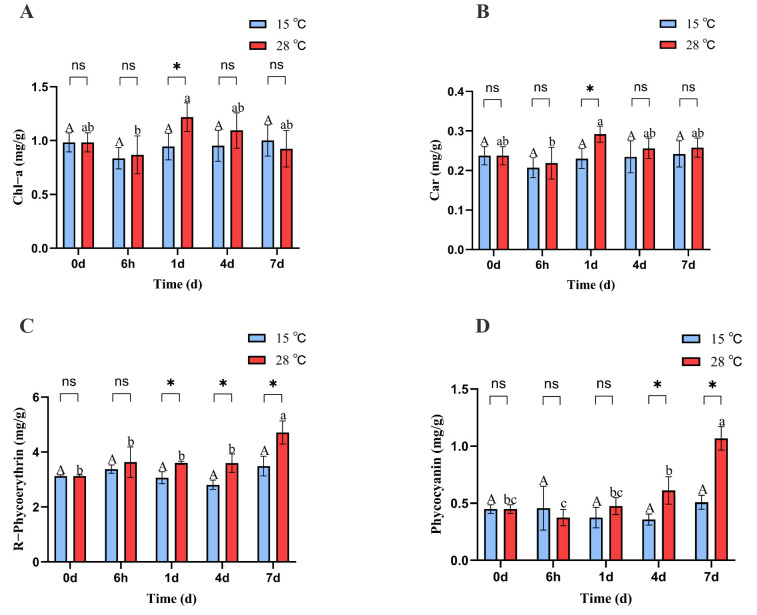
Chl-a, Car, PC, and PE content of *B. fuscopurpurea* at different temperatures. Note: In the figure, A, B, C, D represent the content graphs of chlorophyll, carotenoids, phycocyanin, and phycobilin respectively. The capital letters A, B represent the significant differences in the 15 °C treatment group under different culture times; the lowercase letters a, b, and c represent the significant differences in the 28 °C treatment group under different culture times; ns indicates that there is no significant difference between the 15 °C treatment group and the 28 °C treatment group at the same time point; * represents that there are statistical differences between the 15 °C treatment group and the 28 °C treatment group at the significance level of *p* = 0.05 at the same time point.

**Figure 3 cimb-47-00484-f003:**
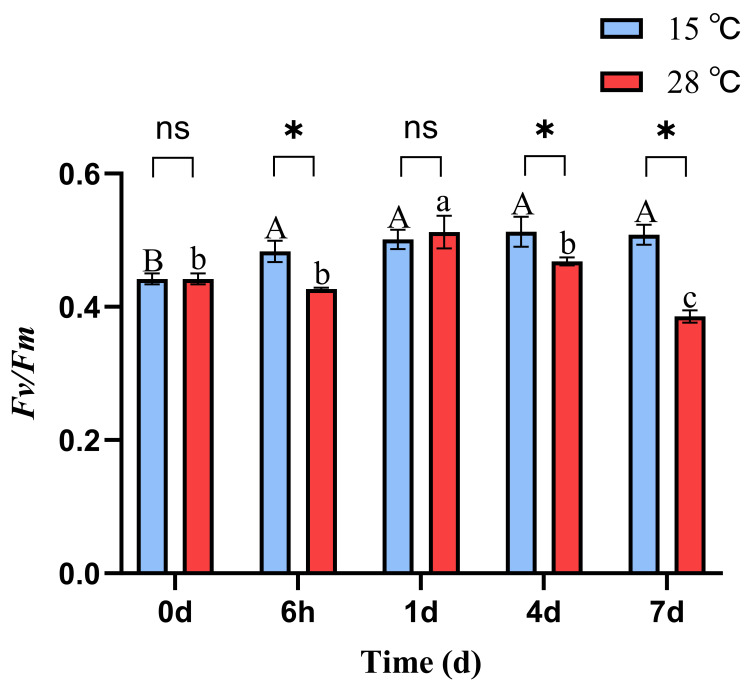
*Fv/Fm* of *B. fuscopurpurea* at different temperatures. Note: The capital letters A, B represent the significant differences in the 15 °C treatment group under different culture times; the lowercase letters a, b, and c represent the significant differences in the 28 °C treatment group under different culture times; ns indicates that there is no significant difference between the 15 °C treatment group and the 28 °C treatment group at the same time point; * represents that there are statistical differences between the 15 °C treatment group and the 28 °C treatment group at the significance level of *p* = 0.05 at the same time point.

**Figure 4 cimb-47-00484-f004:**
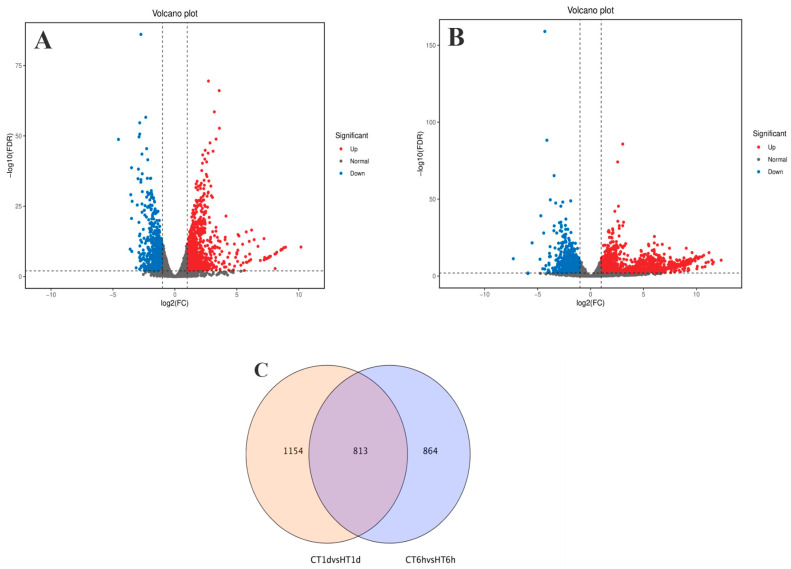
Analysis results of differentially expressed genes between the control group and the high-temperature group (pairwise comparison). Note: (**A**,**B**) are volcano plots of differentially expressed genes in CT group at 6 h compared with HT group at 6 h and CT group at 1 d compared with HT group at 1 d, respectively. The red part represents the number of up-regulated genes, and the blue part represents the number of down-regulated genes. (**C**) is a Venn diagram drawn based on the results of differential expression analysis. The overlapping part refers to the differentially expressed genes that are enriched in both comparison groups. The non-overlapping part refers to the differentially expressed genes that are enriched in each individual group.

**Figure 5 cimb-47-00484-f005:**
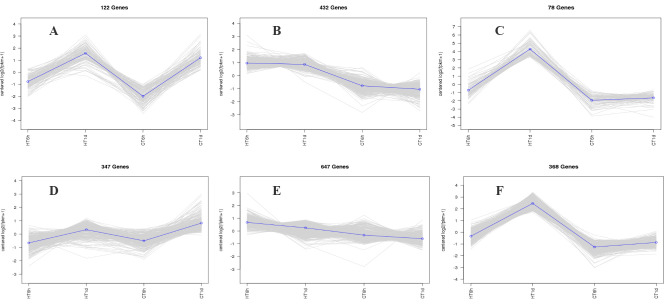
Cluster trend profile chart. Note: The trend clustering curve graph showing similar expression trends for the selected stones (**A**–**F**) in the figure. In this cluster trend profile, the horizontal axis represents different sample treatment groups, and the vertical axis represents the expression trend of the differentially expressed genes that have been enriched. Positive values indicate an upward trend in expression, while negative values indicate a downward trend in expression.

**Figure 7 cimb-47-00484-f007:**
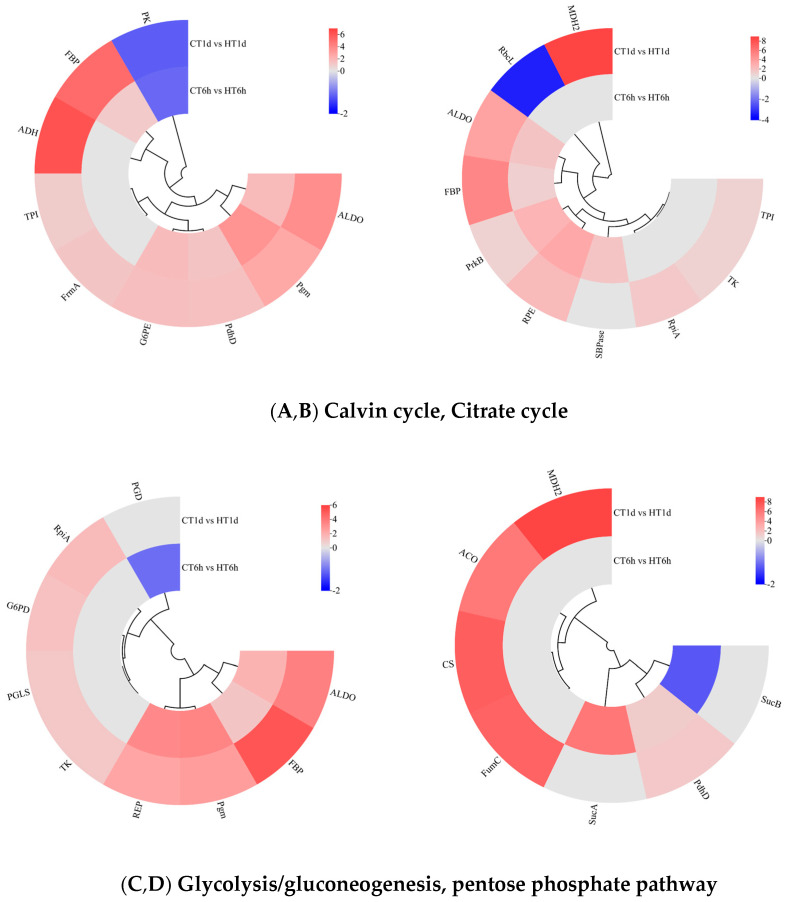
Cluster heatmap of differentially expressed genes in Calvin cycle, Citrate cycle, glycolysis/gluconeogenesis, and pentose phosphate pathway. Note: In the figure, A, B, C, and D represent the cluster heatmaps of the Calvin cycle, Citrate cycle, glycolysis/gluconeogenesis, and pentose phosphate pathway.The logarithmic values of fold change (Log_2_FC) of expression levels among the sample groups are shown in the legend, where red indicates significant up-regulation, blue indicates significant down-regulation, and gray indicates no significant difference.

**Figure 8 cimb-47-00484-f008:**
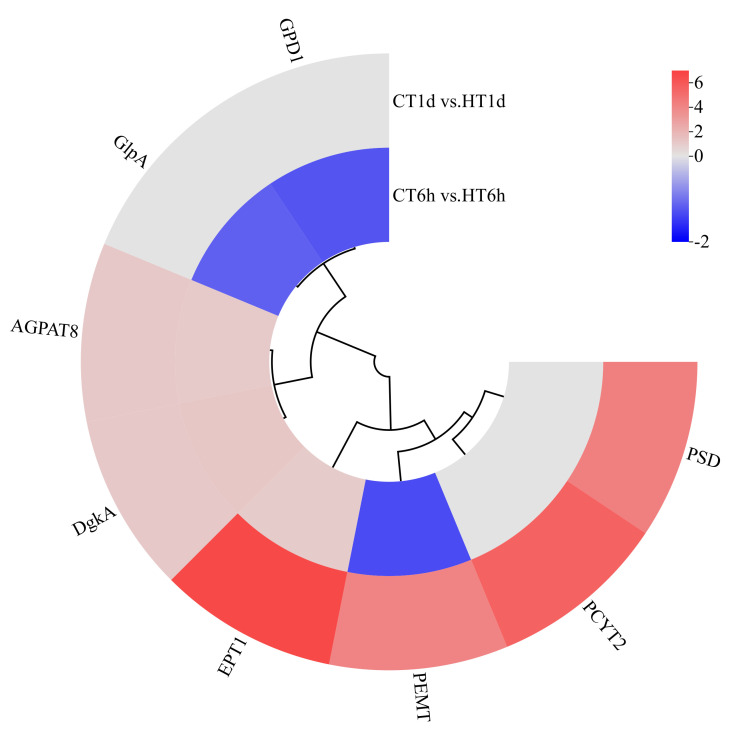
Cluster heatmap of differentially expressed genes in glycerophospholipid metabolism pathway. Note: The logarithmic values of fold change (Log_2_FC) of expression levels among the sample groups are shown in the legend, where red indicates significant up-regulation, blue indicates significant down-regulation, and gray indicates no significant difference.

**Figure 9 cimb-47-00484-f009:**
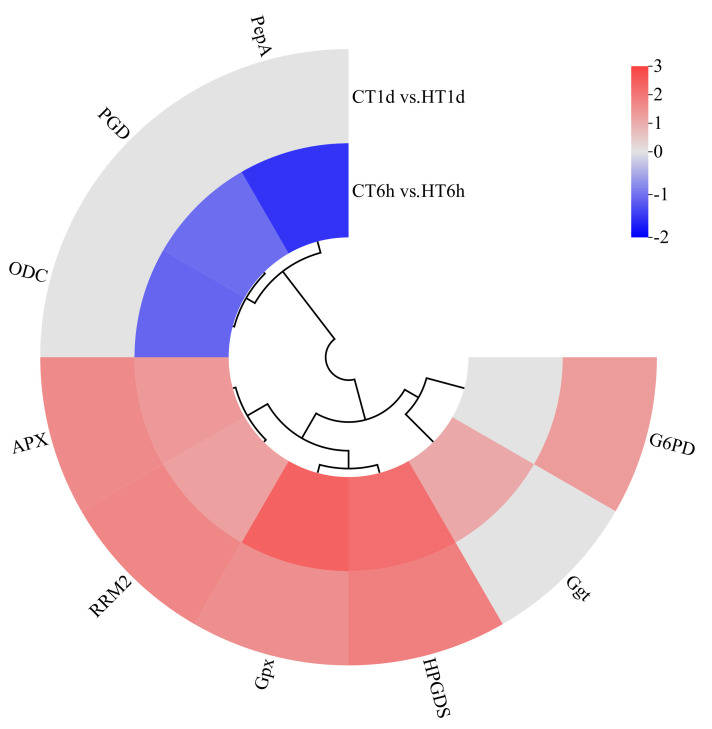
Cluster heatmap of differentially expressed genes in glutathione cycle pathway. Note: The logarithmic values of fold change (Log_2_FC) of expression levels among the sample groups are shown in the legend, where red indicates significant up-regulation, blue indicates significant down-regulation, and gray indicates no significant difference.

**Figure 10 cimb-47-00484-f010:**
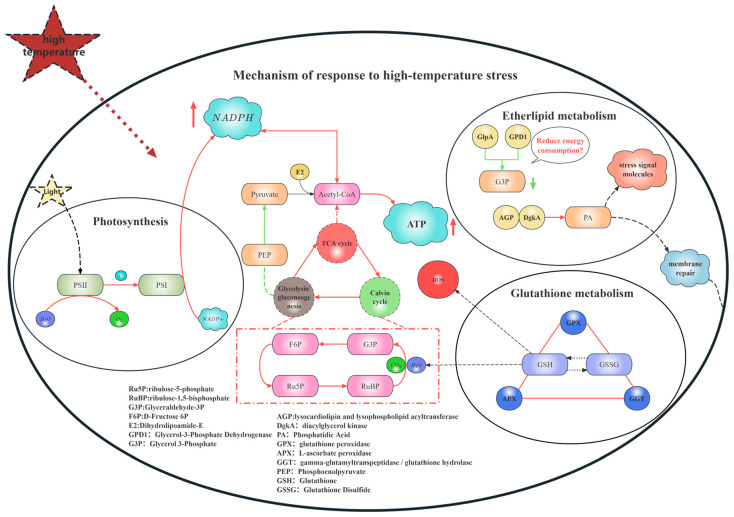
Overall gene regulatory map. Note: In the regulatory diagram of this mechanism, the red arrows indicate that the related regulatory genes are up-regulated, while the green arrows indicate that the related genes are down-regulated.

**Table 1 cimb-47-00484-t001:** Significantly enriched KEGG pathways in CT_6_h vs. HT6h and CT_1_d vs. HT_1_d.

Pathway ID	Pathway	KEGG_B_Class	No. of DEGs	*p* Value
CT6h vs. HT6h				
ko01200	Carbon metabolism	Global and overview maps	29	6.13 × 10^−1^
ko01230	Biosynthesis of amino acids	Global and overview maps	20	6.83 × 10^−1^
ko00860	Porphyrin and chlorophyll metabolism	Metabolism of cofactors and vitamins	19	2.87 × 10^−9^
ko00230	Purine metabolism	Nucleotide metabolism	15	3.25 × 10^−2^
ko00710	Carbon fixation in photosynthetic organisms	Energy metabolism	14	3.65 × 10^−2^
ko00010	Glycolysis/gluconeogenesis	Carbohydrate metabolism	13	2.72 × 10^−1^
ko00630	Glyoxylate and dicarboxylate metabolism	Carbohydrate metabolism	13	3.46 × 10^−1^
ko00196	Photosynthesis antenna proteins	Energy metabolism	12	1.68 × 10^−8^
ko00195	Photosynthesis	Energy metabolism	11	5.94 × 10^−4^
ko00030	Pentose phosphate pathway	Carbohydrate metabolism	11	1.54 × 10^−2^
ko00480	Glutathione metabolism	Metabolism of other amino acids	10	8.50 × 10^−2^
ko00260	Glycine, serine, and threonine metabolism	Amino acid metabolism	10	1.15 × 10^−1^
ko00564	Glycerophospholipid metabolism	Lipid metabolism	9	1.55 × 10^−2^
ko00250	Alanine, aspartate, and glutamate metabolism	Amino acid metabolism	9	1.06 × 10^−1^
CT1d vs. HT1d				
ko01200	Carbon metabolism	Global and overview maps	34	9.43 × 10^−1^
ko01230	Biosynthesis of amino acids	Global and overview maps	29	6.66 × 10^−1^
ko00860	Porphyrin and chlorophyll metabolism	Metabolism of cofactors and vitamins	16	6.43 × 10^−5^
ko00630	Glyoxylate and dicarboxylate metabolism	Carbohydrate metabolism	16	5.52 × 10^−1^
ko00710	Carbon fixation in photosynthetic organisms	Energy metabolism	14	2.87 × 10^−1^
ko00010	Glycolysis/gluconeogenesis	Carbohydrate metabolism	14	6.69 × 10^−1^
ko00195	Photosynthesis	Energy metabolism	13	1.05 × 10^−3^
ko00196	Photosynthesis antenna proteins	Energy metabolism	12	8.25 × 10^−7^
ko00480	Glutathione metabolism	Metabolism of other amino acids	12	1.59 × 10^−1^
ko00230	Purine metabolism	Nucleotide metabolism	12	6.24 × 10^−1^
ko00270	Cysteine and methionine metabolism	Amino acid metabolism	11	6.12 × 10^−1^
ko00030	Pentose phosphate pathway	Carbohydrate metabolism	10	2.14 × 10^−1^
ko00020	Citrate cycle (TCA cycle)	Carbohydrate metabolism	9	9.25 × 10^−1^

## Data Availability

Data is contained within the article or Appendix A.

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
