# Peer review of "Physiological and Biochemical Responses and Transcriptome Analysis of Bangia fuscopurpurea (Rhodophyta) Under High-Temperature Stress"

_cimb, 2025, doi:10.3390/cimb47070484_

Round 1
Reviewer 1 Report
Comments and Suggestions for Authors
This manuscript entitled “Physiological and biochemical responses and transcriptome analysis of Bangia fuscopurpurea (Rhodophyta) under high temperature stress” investigated the effects of heat stress (28℃) on the growth, photosynthesis and transcriptome of Bangia fuscopurpurea. This manuscript presented some new findings, but it needs to be revised in the following aspects:
- In the MM section, “acclimation conditions were set to 15°C”, does the air temperature in the incubator was set at 15°C? or the water temperature was set at 15°C (possibly using a water heater)? Because the air temperature in Fujian, China is usually higher than 15°C, why the authors used such a low temperature as Control in this study? The same issue also occurs in the high temperature treatment (28℃), and it should be better to include other temperature treatment higher than 28℃.
- In the section of 2.5, the authors didn’t provide detailed methods for plant growth and treatment before transcriptome sequencing. At the results section, it seems that they treated the plants with 15°C and 28℃ for 6 h and 1 d.
- There are some errors in Figure 2 and 3. Generally, the highest value in histogram is labled with letter a or A, but the authors labled a or A for the lowest value in the manuscript.
- The layout of some Figures should be revised, such as Fig. 4 and 7. The Figure should be put in one single page.
- The physiological data showed that heat stress affected the content of pigments such as Chla, Car, R-Phycoerythrin and phycocyanin, I think the authors should analyze the DEGs related to the biosynthesis or degradation of these pigments after transcriptome sequencing.
Author Response
Cimments1: In the MM section, “acclimation conditions were set to 15°C”, does the air temperature in the incubator was set at 15°C? or the water temperature was set at 15°C (possibly using a water heater)? Because the air temperature in Fujian, China is usually higher than 15°C, why the authors used such a low temperature as Control in this study? The same issue also occurs in the high temperature treatment (28℃), and it should be better to include other temperature treatment higher than 28℃.
Response1: Thank you for pointing this out. We are using an intelligent light-controlled incubator. The set temperature is 15℃ and 28℃. The measured water temperature deviation does not exceed 0.5℃. In this study, 15℃ was set as the control group because it was determined based on the preliminary experimental results of our research group and the research findings of other domestic research institutions (for example, a paper by Niu was cited in the text). 28℃ was set as the high-temperature stress temperature. The main purpose of this study was to explore the biological changes of B. fuscopurpurea under high-temperature stress.
Cimments2: In the section of 2.5, the authors didn’t provide detailed methods for plant growth and treatment before transcriptome sequencing. At the results section, it seems that they treated the plants with 15°C and 28℃ for 6 h and 1 d.
Response2: Thank you for pointing this out. I have already provided the processing method for the transcriptome experimental materials in the third paragraph of Section 2.1.
Cimments3: There are some errors in Figure 2 and 3. Generally, the highest value in histogram is labled with letter a or A, but the authors labled a or A for the lowest value in the manuscript.
Response3: Thank you for pointing this out. The modifications have been made as per your request. Thank you for your meticulousness.
Cimments4: The layout of some Figures should be revised, such as Fig. 4 and 7. The Figure should be put in one single page.
Response4: Thank you for pointing this out. The modifications have been made as per your request. Thank you for your meticulousness.
Cimments5: The physiological data showed that heat stress affected the content of pigments such as Chla, Car, R-Phycoerythrin and phycocyanin, I think the authors should analyze the DEGs related to the biosynthesis or degradation of these pigments after transcriptome sequencing.
Response5: Thank you for pointing this out. Thank you very much for raising this crucial question. Heat stress does indeed affect the content of photosynthetic pigments. I am also very willing to analyze the differentially expressed genes related to pigment degradation. This is a very good idea. However, unfortunately, when I was writing this paper, my main purpose was to explore the changes in the differentially expressed genes of several main systems of red algae under high-temperature stress, and to draw a preliminary gene regulatory network. Your opinion can be used as my next step of work. Thank you again!
Reviewer 2 Report
Comments and Suggestions for Authors
Dear author,
The manuscript is well written from my point of view. But it needs some minor revision

English Quality can be enriched
Author Response
Cimments1:Introduction Introduction need to arrange by below following sequences 1 st para-Global warming, Heat stress, Plant heat stress, Ocean warming 2 nd para-Importance of Micro algae, Role of Micro algae in mitigating ocean warming, 3 rd para-Photosynthesis in algae, Brief talks on algae Photosynthesis altered by Heat stress 4 th para-Bangia species, China perspective Objectives of this study
Response1: Thank you for pointing this out. The modifications have been made as per your request.
Cimments2: From 2.1-2.7 every section needs specific objectives and more references (make it more reproducible) Little more details is suggested in Statistical analysis “Cluster trend profile chart, pairwise comparison, volcano plot” It is recommended to draw a figure for step by step process of every methodology.
Response2: Thank you for pointing this out.Thank you very much for your suggestion. However, I'm sorry to say that there are relatively few research papers on the large seaweed species known as "red hair algae". My research methods mainly come from studying other domestic research institutions and accumulating experimental experience within my research group. It's difficult to find precise and reliable reference materials. A large part of the method steps were independently explored and developed by me.
Cimments3: It is highly recommended to write the figure caption more self-explanatory with details.
Response3: Thank you for pointing this out. The modifications have been made as per your request.
Cimments4: Well-structured and well written. Ensure 90% references are from 2020.
Response4: Thank you for pointing this out. The modifications have been made as per your request. Thank you for your meticulousness.
Cimments5: Gap of the study need to be added.
Response5: Thank you for pointing this out. I have addressed the shortcomings of this study and the subsequent work to be carried out.
Round 2
Reviewer 1 Report
Comments and Suggestions for Authors
The authors have addressed my previous comments in the revised version, but some mistakes still needs to be corrected as follows:
- Page 2-3, the highlighted text, 0.3-0.5 grams should be 0.3-0.5 g, At 6 hours and 1 day of the experiment should be at 6 h and 1 d of the experiment.
- Page 12, the legend of Fig. 7, (A,B)Calvin cycle、Citrate cycle should be (A,B)Calvin cycle, Citrate cycle.
- In Conclusion section, i think the authors are trying to reply to my previous comment or possibly other reviewers' comment. But in my opinion, the highlighted text in this section should be rewritten or removed.
Author Response
Cimments1: Page 2-3, the highlighted text, 0.3-0.5 grams should be 0.3-0.5 g, At 6 hours and 1 day of the experiment should be at 6 h and 1 d of the experiment.
Response1: Thank you for pointing this out. This was my oversight. I have made the necessary corrections as per your request. Once again, thank you for your meticulousness.
Cimments2: Page 12, the legend of Fig. 7, (A,B)Calvin cycle、Citrate cycle should be (A,B)Calvin cycle, Citrate cycle.
Response2: Thank you for pointing this out. This was my oversight. I have made the necessary corrections as per your request. Once again, thank you for your meticulousness.
Cimments3: In Conclusion section, i think the authors are trying to reply to my previous comment or possibly other reviewers' comment. But in my opinion, the highlighted text in this section should be rewritten or removed.
Response3: Thank you for pointing this out. Thank you for your suggestion. I also think that deletion is a better option. It disrupted my writing structure.